# Characterization of Two Aggressive PepMV Isolates Useful in Breeding Programs

**DOI:** 10.3390/v15112230

**Published:** 2023-11-08

**Authors:** Cristina Alcaide, Eduardo Méndez-López, Jesús R. Úbeda, Pedro Gómez, Miguel A. Aranda

**Affiliations:** ”Del Segura” Centre for Applied Biology (CEBAS), Consejo Superior de Investigaciones Científicas (CSIC), 30100 Murcia, Spain; calcaide@cebas.csic.es (C.A.); fmendez@cebas.csic.es (E.M.-L.); jerodriguez@cebas.csic.es (J.R.Ú.); pglopez@cebas.csic.es (P.G.)

**Keywords:** bright yellow mosaic, coat protein, mutation, PepMV strains, symptom induction, virus evolution

## Abstract

Pepino mosaic virus (PepMV) causes significant economic losses in tomato crops worldwide. Since its first detection infecting tomato in 1999, aggressive PepMV variants have emerged. This study aimed to characterize two aggressive PepMV isolates, PepMV-H30 and PepMV-KLP2. Both isolates were identified in South-Eastern Spain infecting tomato plants, which showed severe symptoms, including bright yellow mosaics. Full-length infectious clones were generated, and phylogenetic relationships were inferred using their nucleotide sequences and another 35 full-length sequences from isolates representing the five known PepMV strains. Our analysis revealed that PepMV-H30 and PepMV-KLP2 belong to the EU and CH2 strains, respectively. Amino acid sequence comparisons between these and mild isolates identified 8 and 15 amino acid substitutions for PepMV-H30 and PepMV-KLP2, respectively, potentially involved in severe symptom induction. None of the substitutions identified in PepMV-H30 have previously been described as symptom determinants. The E236K substitution, originally present in the PepMV-H30 CP, was introduced into a mild PepMV-EU isolate, resulting in a virus that causes symptoms similar to those induced by the parental PepMV-H30 in *Nicotiana benthamiana* plants. In silico analyses revealed that this residue is located at the C-terminus of the CP and is solvent-accessible, suggesting its potential involvement in CP–host protein interactions. We also examined the subcellular localization of PepGFPm2^E236K^ in comparison to that of PepGFPm2, focusing on chloroplast affection, but no differences were observed in the GFP subcellular distribution between the two viruses in epidermal cells of *N. benthamiana* plants. Due to the easily visible symptoms that PepMV-H30 and PepMV-KLP2 induce, these isolates represent valuable tools in programs designed to breed resistance to PepMV in tomato.

## 1. Introduction

Pepino mosaic virus (PepMV) is a positive-sense single-stranded RNA virus that belongs to the genus *Potexvirus* (family *Alphaflexiviridae*). It was first identified in Peru in 1974 [1] in pepino (*Solanum muricatum*) crops, and then it was described in 1999 affecting tomato (*Solanum lycopersicum*) in the Netherlands [2]; since then, it has become pandemic, causing economic losses in tomato crops worldwide [3,4,5]. The PepMV genome is approximately 6.4 kb in length and contains five open reading frames (ORFs), flanked by two untranslated regions, a methylguanosine cap at the 5′ end and a polyadenylated tail at the 3′ end [6]. ORF1 encodes the putative viral polymerase (RdRp), while ORFs 2, 3 and 4 encode the triple gene block (TGB) proteins TGB1, TGB2 and TGB3, which are involved in virus movement [7,8]. TGB1 can suppress RNA silencing [7,9], while TGB2 and TGB3 likely coordinate with TGB1 to form viral replication complexes, as described for potato virus X [10]. ORF5 encodes the capsid protein (CP), which, together with the genomic RNA, form the PepMV virion. The determination of the cryoEM structure of the PepMV CP in virions has allowed for the identification of three major regions in this protein: a flexible N-terminal arm responsible for the side-by-side CP-CP contact within the viral particle, a core region rich in alpha-helixes that contains the RNA-binding pocket domain and a C-terminal extension that protrudes from the core region and is involved in the longitudinal CP-CP contact, forming the inner channel of the virion [11]. The PepMV CP, in addition to its structural role, is required for the cell-to-cell and long distance movements of the virus [7,11], and it is also an RNA silencing suppressor [9]. Potexviral CPs are involved in interactions with host factors [12,13,14,15] and have been associated with symptom induction [16,17]. In the case of PepMV, several host factors have been identified to interact with CP [9,18,19]. Furthermore, specific point mutations within the PepMV genome have been found to be associated with the loss of recognition by the potato *Rx* gene [20,21] or to have an impact on the symptomatology induced by this virus [22,23], with some of these mutations located within the CP coding gene [24,25].

Five PepMV strains have been described thus far, the European (EU), Chilean (CH2), North American (US1/CH1), original Peruvian (LP) and new Peruvian (PES) strains [5,26]. The symptoms caused by PepMV in tomato plants are highly variable and can affect both leaves and fruits, resulting in the irregular pigmentation and discoloration of fruits, leaf bubbling, leaf chlorosis and yellowing and, less often, leaf necrosis [5,22,27]. Symptom severity is influenced by factors such as the virus genotype, tomato cultivar and environmental conditions, including the temperature during host growth [23,28]. The identification of the genetic determinants of symptom induction is not straightforward, as symptom development can be influenced by numerous factors [28]. Nonetheless, certain point nucleotide substitutions in the PepMV genome have been identified as determinants of yellow mosaics or necrosis [23,24,28]. Indeed, two point mutations in the PepMV CP gene have been reported to be associated with the occurrence of interveinal leaf yellowing symptoms; substitutions E155K and D166G, identified in the PepMV CP of different PepMV-CH2 yellowing-inducing isolates, when introduced independently into a mild PepMV-EU or PepMV-CH2 background, reproduced the interveinal leaf yellowing induced by the original isolates [24,29]. In this study, we focused on the description and analysis of PepMV-H30 and PepMV-KLP2, two aggressive isolates collected from commercial tomato fields in South-Eastern Spain, which cause bright yellow mosaics. Full-length infectious cDNA clones were prepared for both. Their phylogenetic relationships were studied, through the inclusion of another 35 full-length genomic sequences from PepMV isolates belonging to the five strains described worldwide. Further molecular analyses led to the identification and in silico characterization of the point substitutions responsible for the induction of yellow mosaics in *Nicotiana benthamiana* plants.

## 2. Materials and Methods

### 2.1. Plant and Virus Materials

Surveys for the identification of bright yellow mosaic-inducing PepMV isolates were carried out during the spring in 2014 and 2015. Tomato crops from the provinces of Granada and Alicante (Spain) were inspected after contacting local growers. Samples consisted of young leaves from recently emerged shoots showing mosaics. In addition to bright mosaics, the plants sampled from Granada showed fruit necrosis, while the plants sampled from Alicante showed severe fruit blotching.

For experimental inoculations, tomato (cv. Moneymaker) and *N. benthamiana* plants were grown in a greenhouse (16 h photoperiod, 26/22 °C day/night cycle) and a growth chamber (16 h photoperiod, 22 °C), respectively. After two weeks post-germination for tomato and post-transplanting for *N. benthamiana*, six plants from each species were inoculated with 2.5 µg of PepMV-Sp13, PepMV-PS5, PepMV-H30 or PepMV-KLP2 purified virions (see below). Symptomatology was assessed 10 days after inoculation. The experiment was repeated thrice.

To identify the point mutation responsible for bright yellow mosaic induction, three *N. benthamiana* plants were agroinfiltrated with *Agrobacterium tumefaciens* liquid cultures transformed with each of the following plasmids: pPepMV-Sp13^A113S^, pPepMV-Sp13^E236K^, pPepMV-Sp13 and pPepMV-H30. Symptomatology was then assessed 10 days after agroinfiltration. The experiment was repeated twice.

To determine the subcellular localization of PepGFPm2, PepGFPm2^A113S^ and PepGFPm2^E236K^, three *N. benthamiana* plants were agroinfiltrated with pPepGFPm2, pPepGFPm2^A113S^ and pPepGFPm2^E236K^ diluted in MES 100 mM, MgCl_2_ 10 mM and acetosyringone 100 mM. The experiment was repeated twice.

### 2.2. Preparation of Infectious cDNA Clones

Total RNA was extracted from the *N. benthamiana* plants inoculated with PepMV-KLP2 or PepMV-H30 using Tri-reagent (Sigma-Aldrich, St. Louis, MO, USA). Retrotranscription was carried out using Reverse Transcriptase (Roche, Basel, Switzerland). Then, cDNAs to the full-length PepMV genomes and the pJL 89 vector were amplified with Phusion high-fidelity DNA polymerase (Thermo Fisher Scientific, Waltham, MA, USA) using strain-specific primers for PepMV-EU or PepMV-CH2 [30] and primers for pJL 89 [31]. The amplified fragments were then purified and cloned into the pJL 89 vector using Gibson Assembly Master Mix (New England Biolabs Inc., Ipswich, MA, USA) [31,32] following the manufacturer’s instructions. Stellar competent *Escherichia coli* cells (Clontech Laboratories, Mountain View, CA, USA) were transformed, and plasmids were sequenced using the Sanger methodology (STAB VIDA, Caparica, Portugal). Finally, the *A. tumefaciens* C58C1 strain was transformed with the purified plasmids.

### 2.3. Virion Purification

Twelve *N. benthamiana* plants were agroinfiltrated with pPepMV-PS5 [33], pPepMV-KLP2, pPepMV-Sp13 [6] or pPepMV-H30. Seven days post-inoculation, the apical leaves were collected and stored at −80 °C before further processing. Frozen samples were ground in liquid N_2_ and homogenized in a buffer containing 0.1 M Tris-citric acid (pH 8.0), 0.2% 2-mercaptoethanol, 0.01 M sodium thioglycolate and 1% Triton X-100 (*v*/*v*). Virion purification was then performed as described by AbouHaidar et al. (1998) [34]. Chloroform was added to the homogenate at a 1:4 ratio. The supernatant was collected after a brief centrifugation, after which an incubation with PEG 8000 and a series of centrifugations were performed. Finally, the pellet was resuspended in 0.1 M Tris-citric acid (pH 8.0), and virion quantification was performed by measuring the optical density at 260 nm and using 2.9 as the extinction coefficient [34].

### 2.4. Comparison of In Silico Sequences

The plasmids pPepMV-H30 and pPepMV-KLP2 were Sanger-sequenced (STAB VIDA) via primer walking using 24 different primers for each isolate to achieve sufficient coverage (Appendix A). Contigs were assembled using Geneious 10.0.9 software. The deduced full genome sequences of PepMV-H30 and PepMV-KLP2 were aligned with another 35 reference PepMV sequences from GenBank, belonging to the EU, CH2, US1, LP and PES strains. SDT v1.2 software was used to compute pairwise sequence identities between the different sequences [35]. The evolutionary history was inferred using MEGA software version 7 [36], employing the Maximum Likelihood method based on the General Time Reversible model. Synonym and non-synonym substitutions were assessed between the sequences to identify differences and determine the point substitutions that may be potentially involved in the development of bright yellow mosaics.

### 2.5. Structural Modeling of the PepMV-H30 CP

The PepMV-H30 CP was modeled de novo using the I-TASSER tool [37,38,39]. The model with the highest C-score was used for further analyses. The 3D representations and structural alignments of proteins were generated using PyMol software version 2.5 (Schrödinger, New York, NY, USA). The estimation of the electrostatic potential of the surface of the CP was carried out using the APBS PyMol plugin [40]. Complexes built by 6 CP subunits (N_i_, N_i+1_, N_i+3_, N_i+9_, N_i+10_, N_i+11_), as well as the viral RNA, were generated via structural alignment using the cryoEM model of the PepMV particle (PDB code 5fn1) [11] as the template. Polar contacts and hydrogen bond networks between the residues of loose CPs or the 6-subunit CP complexes were predicted using PyMol and the *ProteinTools* online server [41], respectively.

### 2.6. Site-Directed Mutagenesis

Two single-nucleotide substitution mutants of PepMV-Sp13 were constructed, with each encoding one non-synonymous substitution in the CP coding region, A113S or E236K; the substitutions were 5969G > T or 6338G > A [6]. The PepMV-Sp13 cDNA clone [6] was used as the template, and site-directed mutagenesis was performed using a CloneAmp HiFi PCR premix (Takara, Japan) with specific primers for PepMV-Sp13^A113S^: CE-2915 5′-*cgccgagccctttctgctcagtttg*-3′ and CE-2916 5′-*caaactgagcagaaagggctcggcg*-3′ and for PepMV-Sp13^E236K^, CE-2917 5′-*gacgcaccccctaaactttaaacac*-3′ and CE-2918 5′-*gtgtttaaagtttagggggtgcgtc*-3′. The amplified products were digested with DpnI (New England Biolabs, Ipswich, MA, USA) and transformed into competent *E. coli* cells. Then, Sanger sequencing (STAB VIDA, Portugal) was used to confirm that each mutation was successfully incorporated. These DNA constructs were used to assess the symptomatology associated with each point mutation. Additionally, the same point mutations were introduced into a PepMV-Sp13 clone labeled with GFP using the same process and primers. For that, the PepGFPm2 clone was used as the template [42] to produce the clones PepGFPm2^A113S^ and PepGFPM2^E236K^. In all cases, the GFP gene was linked to the PepMV CP gene through the foot-and-mouth disease virus (FMDV) 2A catalytic peptide. The pGWB452 vector expressing the GFP was used as the negative control.

### 2.7. Confocal Laser Scanning Microscopy

Small pieces of agroinoculated leaves were cut and mounted onto glass microscope slides and observed with a Leica STELLARIS 8 inverted confocal microscope. The scanning was performed by using a 63× magnification glycerol immersion lens and an excitation wavelength of 488 nm for GFP.

## 3. Results

### 3.1. Characterization of Two PepMV Isolates Inducing Severe Bright Yellow Mosaics

We collected samples from tomato plants showing bright yellow mosaics in leaves, along with necrosis in fruits and stems, and marked fruit blotching, from crops located in Granada and Alicante (Spain), respectively. When the samples were analyzed against a panel of viruses known to affect tomato, only PepMV was detected. Four samples from each location were used to inoculate plants of the experimental host *N. benthamiana*, resulting in symptoms that were much more aggressive than those caused by the reference isolate PepMV-Sp13 [6], and these symptoms included bright yellow mosaics in leaves, as in the original tomato host, with these being very homogenous within plants from each location. Two single *N. benthamiana* plants inoculated with a sample from each location were selected as the sources of virus inocula, and they were designated PepMV-KLP2 (Granada) or PepMV-H30 (Alicante). RNA was extracted from the *N. benthamiana* plants and used to molecularly clone PepMV. cDNA to the full-length PepMV genome was prepared, first cloned into an *E. coli* vector and then transferred into a binary vector. The agroinoculation of the *N. benthamiana* plants with four clones per isolate reproduced the aggressive symptoms observed in plants of this species for all the assayed clones. Therefore, the decision was made to continue with one clone per location, maintaining the original names, PepMV-KLP2 for the clone from Granada and PepMV-H30 for the clone from Alicante. Next, we purified virions from the *N. benthamiana* plants agroinoculated with each of the clones and used them to inoculate tomato plants. In this experiment, we included PepMV-Sp13 (EU strain) [6] and PepMV-PS5 (CH2 strain) [33] as the references; both have been described to induce mild symptoms in tomato [43]. Tomato plants grown under experimental greenhouse conditions reproduced the symptomatology observed in infected commercial tomato crops. For PepMV-Sp13 and -PS5, we observed mild, almost inconspicuous symptoms in leaves (Figure 1A,B), while for PepMV-H30 and PepMV-KLP2, we observed bright yellow mosaic, vein banding, chlorosis and leaf distortion (Figure 1C–F).

The full-length sequences of the PepMV-H30 and PepMV-KLP2 genomes were determined via Sanger sequencing and compared with another 35 sequences obtained from the GenBank database corresponding to isolates belonging to the five PepMV strains described, LP, EU, US1, CH2 and PES. The 37 full-length nucleotide sequences were used to calculate pairwise sequence identities. Higher similarities were found between the EU and LP isolates (95%), and between the US1 and PES isolates (87%). The EU and LP isolates showed about 83% similarity with the US1 and PES isolates and about 80% with the CH2 isolates. The largest differences were found between the CH2 and US1 and PES isolates (79%) (Figure 2A) (Appendix A). A phylogenetic analysis (Figure 2B) showed three clear clades, one composed of PES and US1 isolates, another composed of CH2 isolates and a third one composed of LP and EU isolates. Upon these sequence analyses, we classified PepMV-H30 as belonging to the EU strain, whereas PepMV-KLP2 was categorized under the CH2 strain. As far as we are aware, this is the first time that a natural PepMV isolate belonging to the EU strain has been shown to induce bright yellow mosaics.

### 3.2. Newly Described Mutations Associated with Yellow Mosaic Induction in N. benthamiana

The nucleotide and amino acid sequences of the PepMV-H30 and PepMV-KLP2 isolates, respectively, were compared with those of the well-characterized PepMV-Sp13 and PepMV-PS5 isolates. In the case of PepMV-H30, we found 14 and 8 nucleotide and amino acid differences, respectively, with respect to PepMV-Sp13. In the case of PepMV-KLP2, we found 78 and 15 nucleotide and amino acid substitutions, respectively, in comparison with PepMV-PS5. Since the mutations in the CP ORF have been identified as being responsible for bright yellow mosaic induction in other PepMV isolates [24], we focused on the substitutions in this genomic region as possible symptom determinants. We found two point substitutions in the PepMV-KLP2 CP sequence, one encoding a change from glutamic acid to lysine in residue 155 (E155K) and the other encoding a change from serine to phenylalanine in residue 94 (S94F). For the PepMV-H30 CP sequence, we found two non-synonymous point mutations, one encoding the substitution of alanine to serine in position 113 (A113S) and the other encoding the substitution of glutamic acid to lysine in position 236 (E236K). The E155K substitution found in PepMV-KLP2 has previously been described as responsible for bright yellow mosaic induction [24], but none of the mutations identified in PepMV-H30 have been reported before as symptom determinants. Therefore, we focused on the latter, and through directed mutagenesis, we introduced each point mutation separately into the genome of PepMV-Sp13 to study symptom induction. We then compared the symptomatology induced by PepMV-Sp13, PepMV-H30, PepMV-Sp13^A113S^ and PepMV-Sp13^E236K^ in the *N. benthamiana* plants (Figure 3). No yellow mosaic symptoms were observed in the plants infected with PepMV-Sp13^A113S^ (Figure 3B), similar to PepMV-Sp13-infected plants (Figure 3C). In contrast, yellow mosaic symptoms were observed in PepMV-Sp13^E236K^-infected plants (Figure 3D), similar to those found after PepMV-H30 infection (Figure 3E). Therefore, substitution E236K in the PepMV CP appeared to be a determinant of yellow mosaic induction in *N. benthamiana* plants, at least when expressed from the EU genome background.

Next, we analyzed in silico the effect of the amino acid substitutions on the structure of the PepMV-H30 CP. Both the atomic model of the virion and the CP of the PepMV-Sp13 isolate are available in the Protein Data Bank (PDB) under accession number 5fn1 [11]. The structure of PepMV-H30 was predicted de novo using the I-TASSER server [37,38,39], and the model obtained had a confidence score (C-score) of −0.50. The alignment of the PepMV-Sp13 and PepMV-H30 CP models showed a root-mean-square deviation (RMSD) of 0.797 Å (Appendix A), indicating that both structures are very similar and that the substitutions A113S and E236K do not dramatically affect the protein structure. Alanine and serine are both small amino acids; alanine is hydrophobic, and serine is hydrophilic, but both share similarities. Therefore, the substitution is considered a conservative replacement [44]. Glutamic acid and lysine are negatively and positively charged amino acids, respectively, and this substitution is a non-conservative replacement [44]. Residue A113 is located in the core region of the protein, inside a groove with a high electropositive potential, whereas E236 is the penultimate residue of the protein located in its C-terminal extension, readily accessible to the solvent (Appendix A and Figure 4) and possibly more prone to establish interactions with host factors.

### 3.3. Subcellular Localization of PepMV-EU Carrying Different Point Mutations in the CP

In order to study the subcellular localization of PepMV-Sp13^E236K^, we introduced the mutation responsible for the E236K substitution into the PepGFPm2 infectious clone, which is based on PepMV-Sp13 and expresses the GFP fused to the CP through the 2A catalytic peptide [42]. We have successfully used PepGFPm2 to identify and study the PepMV cellular factories in the past [18,42,45]. We also prepared PepGFPm2^A113S^, which, together with the original PepGFPm2, were the controls of this experiment. We thus agroinfiltrated the *N. benthamiana* plants with each of the PepGFPm2^E236K^, PepGFPm2^A113S^ and PepGFPm2 variants, and, after three days, fluorescence emission was visualized using confocal laser scanning microscopy. We found similar GFP subcellular distributions in the plants infected with all three virus variants, and no GFP fluorescence was observed in chloroplasts or in their proximity for any of them (Figure 5).

## 4. Discussion

Generating knowledge on the genetic determinism of aggressive symptoms is fundamental for forecasting and preventing damaging epidemics. For PepMV, it has been shown that point mutations in the TGB3 or CP coding genes are linked to symptom aggressiveness, including severe mosaics or necrosis [22,23,24]. In this work, we characterized PepMV-H30 and PepMV-KLP2, two isolates belonging to the EU and CH2 strains, respectively. In their genomes, we identified mutations potentially causative of bright yellow mosaics in plants. The mutations in the PepMV-H30 CP have never been described before. Therefore, a decision was made to focus on these. Other authors have associated chlorosis and yellowing symptoms to the localization of the corresponding viruses in chloroplasts [46,47,48]. For instance, an early work showed that a chlorosis-inducing isolate and a mild isolate of tobacco mosaic virus (TMV) localized to chloroplasts in different amounts and that the chlorosis-inducing isolate was able to inhibit photosystem II activity [47]. PepMV forms its factories in association with the ER [42]; thus, we hypothesized that the E236K substitution in the PepMV CP could cause a change in virion localization, presumably to chloroplasts. However, our analyses using GFP-tagged viruses did not show differences in the localization of the GFP fluorescence between the mutant and wild-type viruses. As a cautionary note for the interpretation of our results, wild-type and mutant PepMV clones were overexpressed in the *N. benthamiana* plants, and microscopy observations corresponded to epidermal cells; therefore, it is possible that, in mesophyll cells and/or in tomato plants, the localization of the wild-type and mutant viruses differs, differentially affecting chloroplasts. Chloroplasts are a common target during virus infections, and, at the same time, this organelle plays important roles in plant defense [49]. It is possible that, in the case of PepMV, even if there is no association between localization and symptomatology, an indirect effect may be perturbing chloroplast functioning, but this is a hypothesis that awaits further consideration. Another question is related to the mechanisms that trigger these effects. We analyzed the electrostatic potential of both PepMV-Sp13 and PepMV-H30 CPs, and we observed potential electrostatic changes on the overall surface of the H30 CP (Figure 4). A113 may participate in the stabilization of the alpha-helix and the longitudinal CP-CP contact within the virion. The Ala to Ser replacement in position 113 may have an effect on this CP-CP interacting interface. The replacement of Glu to Lys in position 236 may affect the contact network in the virion inner channel, as both amino acids are differently charged, and the electrostatic potential of the C-terminal extension is predicted to be less negative in the PepMV-H30 CP model (Figure 4). Unfortunately, the resolution of the PepMV CP cryoEM model drops at the C-terminal segment of the C-terminal extension, making a more in-depth structural analysis unreliable. However, in general terms, the substitutions did not affect the overall folding of the CP or the structure of the protein in a decisive way, and the fact that they are solvent-accessible strongly suggests that these substitutions may affect the CP interaction with (a) host factor(s). Interactions between the PepMV CP and host proteins have been shown to have important effects on virus biology, e.g., [18].

Viruses exhibit high mutation rates [50], which may or may not result in high variability and a rapid evolution of virus populations, e.g., [51,52]. The viruses’ mutational propensity has detrimental effects, but it provides the plasticity that allows viruses to rapidly evolve and adapt to environmental changes, e.g., [53,54]. In the case of PepMV, sequential epidemic outbreaks have been associated with the appearance of different viral strains [33,55], but the PepMV CH2 strain has been largely predominant in many geographical areas, including in Spain [30,56], Poland [57], Belgium [58] and North America [59], and no recent displacement of variants has been reported. This is in spite of the description of PepMV isolates inducing particularly aggressive symptoms, for instance, fruit necrosis and/or severe mosaics [22,23,24], in this work. It is generally assumed that aggressive symptoms are associated with increased within-host virus accumulation, and, indeed, the accumulation of PepMV-KLP2 has been reported to be higher than that of other CH2 isolates [60]. However, virus accumulation has been positively correlated with virus transmission in different virus species [61,62,63], including PepMV [64]. Therefore, the limited spread of aggressive isolates such as PepMV-KLP2 or PepMV-H30 could be explained by appealing to the virulence–transmission trade-off hypothesis [65,66]. Counter-selecting factors may operate at the landscape level, but within-host factors may also select against them; it has been suggested that mutations that affect the TGB1 and CP PepMV genes have generally deleterious effects [29]. Hasiów-Jaroszewska et al. [22] also suggested that selection pressure could act to the disadvantage of yellowing-producing PepMV genotypes, as interveinal leaf yellowing symptoms gradually disappeared after several weeks [24]. In addition, in an experimental evolution assay, it was demonstrated that the aggressiveness of PepMV-KLP2 decayed rapidly after passaging, with plants showing mild or no symptoms after three passages [43]. Therefore, while the emergence of isolates such as PepMV-H30 or -KLP2 may occur occasionally, their spread in tomato crops may be rare.

From a very different perspective, the isolates PepMV-H30 and -KLP2 represent valuable tools for breeding resistance against PepMV. In breeding programs, an early selection of resistant/tolerant plants is often required [67], but virus-induced symptoms might not always be easily distinguishable. Inoculations with isolates that induce very obvious symptoms, such as those induced by PepMV-H30 and -KLP2, may simplify the breeders’ work, particularly when large collections of plants need to be characterized. Indeed, PepMV-H30 and -KLP2 have been successfully used recently for the screening of a large collection of tomato mutants [60].

## Figures and Tables

**Figure 1 viruses-15-02230-f001:**
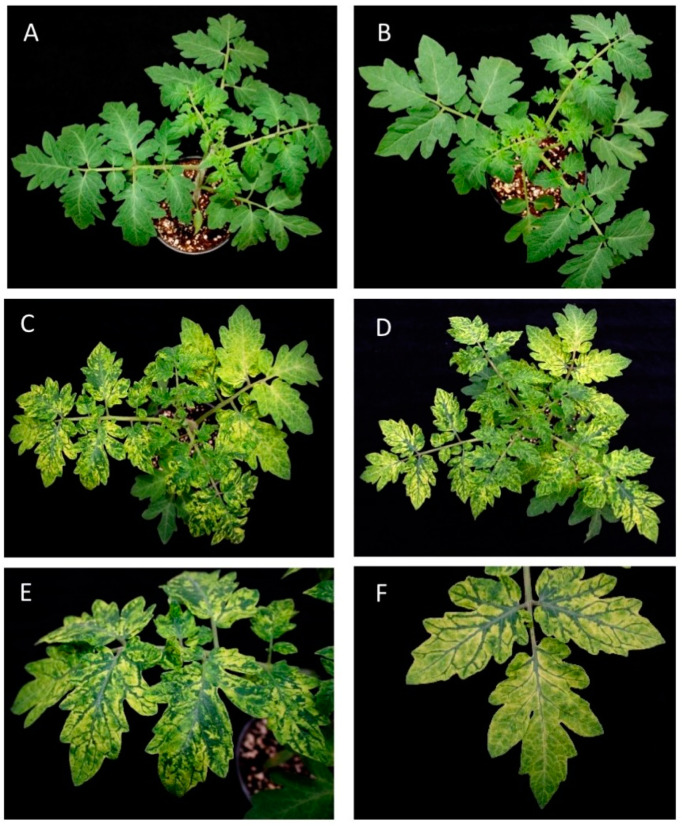
Symptoms induced in tomato plants by different PepMV isolates. (**A**) Tomato plants infected with a PepMV-EU isolate (PepMV-Sp13). (**B**) Tomato plants infected with a PepMV-CH2 isolate (PepMV-PS5). (**C**) Tomato plants infected by an aggressive PepMV-EU isolate (PepMV-H30). (**D**) Tomato plants infected by an aggressive PepMV-CH2 isolate (PepMV-KLP2). (**E**) Detail of (**C**). (**F**) Detail of (**D**). In plants infected by aggressive isolates, we can observe intense bright yellowing in tomato leaves, whilst in plants infected by mild isolates, no symptoms are observed.

**Figure 2 viruses-15-02230-f002:**
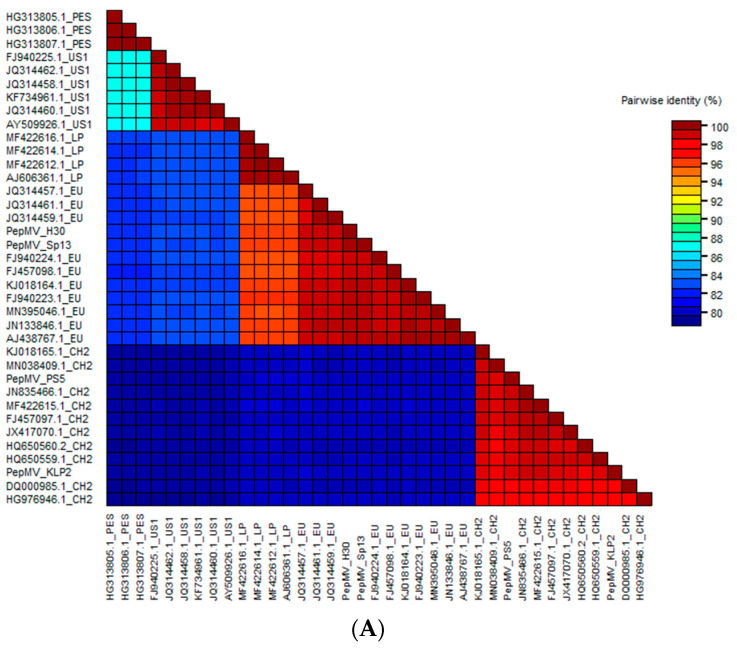
(**A**) Pairwise identity matrix for 37 PepMV nucleotide sequences. Each cell represents an identity score between two sequences. GenBank accession numbers are shown together with the strain to which each isolate belongs. (**B**) Phylogenetic relationships between different PepMV isolates. PepMV-H30, PepMV-KLP2, PepMV-Sp13, PepMV-PS5 and other PepMV reference isolates belonging to the EU, CH2, LP, PES and US1 strains were used to construct the phylogenetic tree. The evolutionary history was inferred using the Maximum Likelihood method. The tree is drawn to scale, with branch lengths measured as the number of substitutions per site. The PES isolates are labeled in grey, the US1 isolates in red, the CH2 isolates in green, the LP isolates in yellow and the EU isolates in blue.

**Figure 3 viruses-15-02230-f003:**
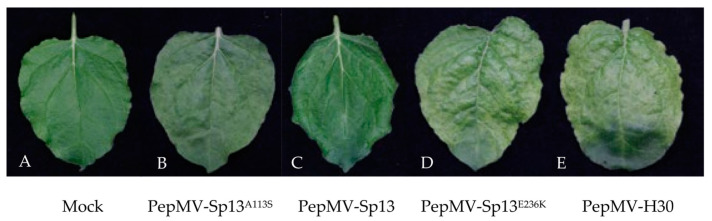
Symptomatology produced in *Nicotiana benthamiana* leaves by different PepMV variants. Samples were taken from (**A**) mock-inoculated plants, (**B**) PepMV-Sp13^A113S^-infected plants, (**C**) PepMV-Sp13-infected plants, (**D**) PepMV-Sp13^E236K^-infected plants, (**E**) PepMV-H30-infected plants. Yellowing can be clearly observed in PepMV-H30- and PepMV-Sp13^E236K^-infected leaves.

**Figure 4 viruses-15-02230-f004:**
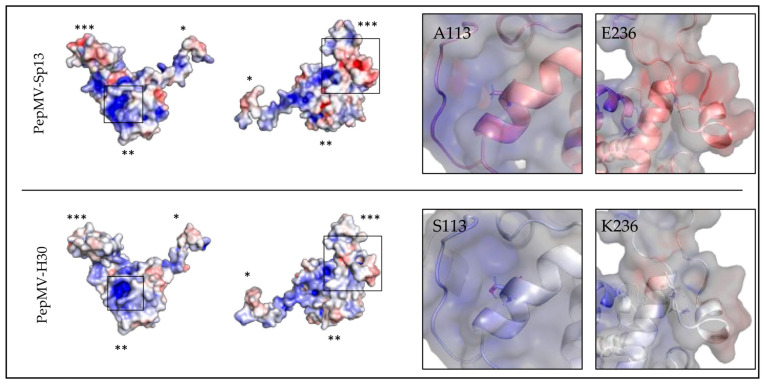
Structural analysis of the effect of the PepMV-H30 CP substitutions. The left side of the panel shows the models of PepMV-Sp13 (PDB code 5fn1) and PepMV-H30 CPs in two different orientations. Molecular surfaces are colored according to their electrostatic potential. The color scale ranges from −5000 T (red) to +5000 T (blue). Furthermore, 1 (*), 2 (**) or 3 (***) asterisks indicate the N-terminal arm, the core region or the C-terminal extension of the CP, respectively. Regions where the mutated residues are located in the PepMV-H30 CP are framed. Close-up views of the framed regions are shown on the right side of the panel. CP surfaces are shown in semi-transparent mode and overlapping the CP atomic models in cartoon mode. Each residue of interest is represented by sticks and centered on the image.

**Figure 5 viruses-15-02230-f005:**
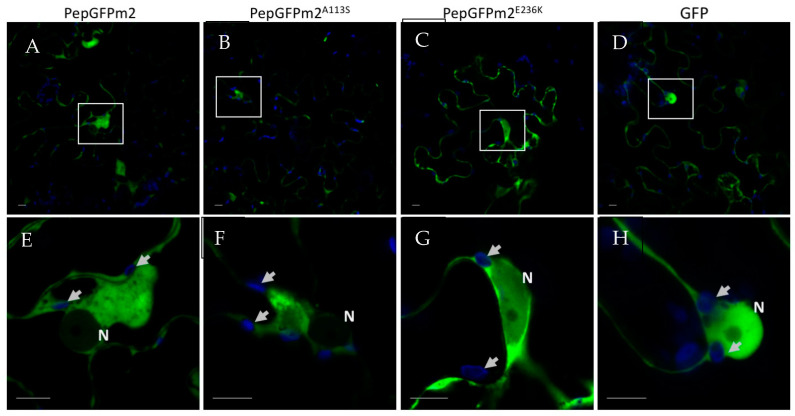
Subcellular localization of the CP of PepMV in epidermal cells of *Nicotiana benthamiana* leaves. Plants were agroinfiltrated with pPepGFPm2, pPepGFPm2^A113S^, pPepGFPm2^E236K^ and pGWB452-GFP to express (**A**,**E**) PepGFPm2, (**B**,**F**) PepGFPm2^A113S^, (**C**,**G**) PepGFPm2^E236K^ and (**D**,**H**) GFP, respectively. Images were taken 3 days after agroinfiltration. Lower panels are magnifications of the upper panels, with the magnified portion indicated by a white empty square in the upper panels. The blue channel highlights the chloroplasts (arrows), while the green channel displays the CP of PepMV, except for in panels D and H, where free GFP is shown as the negative control. N, nucleus. Scale bars correspond to 10 µm.

## Data Availability

PepMV-KLP2 and PepMV-H30 nucleotide sequences have been deposited in GenBank under the accession numbers OR733204 and OR733205, respectively.

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
