# Peer review of "Characterization of Two Aggressive PepMV Isolates Useful in Breeding Programs"

_viruses, 2023, doi:10.3390/v15112230_

Round 1

Reviewer 1 Report

Comments and Suggestions for Authors

The article of Alcaide et al., identifies and characterize two aggressive PepMV isolates (PepMV-H30 and PepMVKLP2) collected in South-eastern Spain. Infectious clones of both isolates were produced, the complete nucleotide sequences were determined and the analysis revealed two amino acid alteration of the cp region. Previously the E155K mutation identified in the PepMV-KLP2 strain was identified as responsible for bright yellow mosaic induction (Hasiów-Jaroszewska et al., 2013, Reference 24), so the other isolate was characterized in detail. In the PepMV-H30 strain two point mutation was identified in the CP region (A113S, E236K), and the E236S was found to induce more severe symptoms. Unfortunately the authors analyzed the symptoms on N. benthamiana plants, but on this host the symptom difference between the usual strains and this strains are not so distinct (Fig 3) as in the case of tomato (Fig 1). So I think that the analysis of the symptoms on tomato plants is essential, since just in this case can be proved that the E236K aa alteration is responsible for the bright yellow symptom induction. I also suggest to change the text (ln255-260), since on N. benthamiana plants “bright yellow mosaic symptoms” are not detected, according to the Fig 3 mild mosaic (Fig 3 B, C ) and more severe mosaic (Fig 3 D, E) can be observed.

The section of analyzing the effect of aa substitution on the coat protein structure in silico is interesting and well documented.

The next section deals with the intracellular localization of the CP determined by confocal microscopy of GFP tagged CP, but again the authors used N. benthamiana plants instead of tomato. Difference was not identified in the intracellular localization of the wild type CP and the CP-E236K. I think that this is not surprising, since on this host just mild symptom difference can be recognized. The analysis of the intracellular localization of these fusion proteins in tomato  mesophyll cells is essential, especially because the authors compare their results to the results of Minicka et al (2015, reference 45) where clear differences were recognized in the intracellular localization of the CP of the wild type and the yellow symptom inducing strain, but they used tomato plants in their experiments, which is more reliable in this respect.

Generally the topic of the article is interesting, the article is well written, but it is essential to repeat the inoculation and intracellular localization experiments on tomato plants.

Author Response

Reviewer #1

In the PepMV-H30 strain two point mutation was identified in the CP region (A113S, E236K), and the E236S was found to induce more severe symptoms. Unfortunately the authors analyzed the symptoms on N. benthamiana plants, but on this host the symptom difference between the usual strains and this strains are not so distinct (Fig 3) as in the case of tomato (Fig 1). So I think that the analysis of the symptoms on tomato plants is essential, since just in this case can be proved that the E236K aa alteration is responsible for the bright yellow symptom induction. I also suggest to change the text (ln255-260), since on N. benthamiana plants “bright yellow mosaic symptoms” are not detected, according to the Fig 3 mild mosaic (Fig 3 B, C ) and more severe mosaic (Fig 3 D, E) can be observed. 

  • It is true that the yellow mosaic was not as bright in N. benthamiana as in tomato, but it was clearly there. Capturing these kind of symptoms in photographs is difficult, and indeed we were only partially able of doing it properly in spite of the many attempts. Also, the quality of the photograph in the pdf for the referees assessment may not be as good as the original (and hopefully the final) one. In any case, we have taken out “bright” when describing the symptoms in N. benthamiana, but maintained “yellow mosaic”. Regarding the comment on testing the mutant in tomato plants, the referee is right. Unfortunately, these are experiments that we have not done, and that we cannot do in the short time available to respond to referees. Therefore, in the new version of the manuscript, we have specified the host along the text when referring to symptoms and avoided generalizations.

The section of analyzing the effect of aa substitution on the coat protein structure in silico is interesting and well documented.

  • Thank you for your encouraging comment.

The next section deals with the intracellular localization of the CP determined by confocal microscopy of GFP tagged CP, but again the authors used N. benthamiana plants instead of tomato. Difference was not identified in the intracellular localization of the wild type CP and the CP-E236K. I think that this is not surprising, since on this host just mild symptom difference can be recognized. The analysis of the intracellular localization of these fusion proteins in tomato  mesophyll cells is essential, especially because the authors compare their results to the results of Minicka et al (2015, reference 45) where clear differences were recognized in the intracellular localization of the CP of the wild type and the yellow symptom inducing strain, but they used tomato plants in their experiments, which is more reliable in this respect. 

  • Thank you very much for your comment. As said before, we have not tested the E236K mutant in tomato, neither to check symptom induction nor for cellular analyses. We have tried to incorporated the referee comments to the new version of the manuscript.  

Reviewer 2 Report

Comments and Suggestions for Authors

In this study, the authors characterized two aggressive PepMV isolates, H30 and KLP2, belonging to the EU and CH2 strains, respectively, that cause severe bright yellow mosaic symptoms in tomato. The E236K substitution in the coat protein (CP) of PepMV-H30 was first noted to be associated with symptom severity compared with mild PepMV-EU isolates. In silico analyses suggested that residue 236 may be involved in the interactions between CP and host factors. Finally, the authors proposed that PepMV-H30 and PepMV-KLP2 cause significant symptoms and can be used as tools for PepMV resistance breeding in tomato. To be honest, I find less novel significance in this study. Although a new amino acid residue in PepMV CP is shown to be associated with symptoms, its effects on the virus, such as replication and movement, have not yet been addressed. I recommend that the authors provide molecular evidence for quantitative detection of viruses in plants to allow comparisons of mild and aggressive isolates. My other concerns are as follows.

1.     P2, L83: Clarify what ‘protected’ tomato crops are. Were the aggressive PepMV isolates selected from the protection of mild PepMV isolates?

2.     P2, L92: The description of PepMV inoculation needs to be clarified. I am not sure if I misunderstood the meaning. To my knowledge, mechanical inoculation of viruses can conventionally be done using plant sap as inocula instead of virions.

3.     P2, L96: Which plant species was/were used for agroinfiltration? N. benthamiana?

4.     P3, L104-114: Add a proper title to this paragraph.

5.     P4, L165: the PepMV CP gene

6.     P5, L215: Numerical values for pairwise sequence identities can be shown in the Supplement for the reader’s reference.

7.     P7, L237: PepMV-Sp13 and PepMV-PS5 isolates, respectively

8.     Fig. 3: The phenotypes of the mutants PepMV-Sp13A113S, and PepMV-Sp13E236K should be illustrated in tomato.

9.     Fig. 5: The locations indicated by boxes should be explained. Organelles such as chloroplasts and nuclei can be indicated.

Comments on the Quality of English Language

No comment.

Author Response

Although a new amino acid residue in PepMV CP is shown to be associated with symptoms, its effects on the virus, such as replication and movement, have not yet been addressed. I recommend that the authors provide molecular evidence for quantitative detection of viruses in plants to allow comparisons of mild and aggressive isolates.

  • Thank you for your comment. PepMV-KLP2 and PepMV-H30 were used to characterize a tomato mutant as described in Ruiz-Ramón et al. (2023). In that report, we showed data on the accumulation of both isolates in susceptible and resistant tomato genotypes as measured by RT-qPCR. Interestingly, accumulation trends were essentially irrespective of the severity of symptoms induced by PepMV-Sp13 or PepMV-H30 (Ruiz-Ramón et al., 2023).

My other concerns are as follows.

  1. P2, L83: Clarify what ‘protected’ tomato crops are. Were the aggressive PepMV isolates selected from the protection of mild PepMV isolates?
  • “Protected” meant net and greenhouse crops. “Protected” is a widely used term to describe both. Perhaps the referee was thinking on cross-protection. If this were case, the answer is no, crops were not under cross-protection schemes. In any case, we have taken out the word “protected” as it does not provide any information relevant to the description or discussion of results.
  1. P2, L92: The description of PepMV inoculation needs to be clarified. I am not sure if I misunderstood the meaning. To my knowledge, mechanical inoculation of viruses can conventionally be done using plant sap as inocula instead of virions.
  • The referee is right, PepMV can be readily transmitted by mechanical inoculation using crude sap from infected plants. Since virion purification is an easy procedure for us, we routinely use purified virion for inoculations for the characterization of isolates and other plant/virus interaction studies. This practice provides more reproducible results, as the virus in the inocula is accurately quantified and we can use always the same amount. Many of our works have been done inoculating purified virion, even if this was not strictly necessary. To clarify, we have added the term “purified”.
  1. P2, L96: Which plant species was/were used for agroinfiltration?  benthamiana?
  • Yes, clarified.
  1. P3, L104-114: Add a proper title to this paragraph.
  • Done, thank you.
  1. P4, L165: the PepMV CP gene
  • Corrected.
  1. P5, L215: Numerical values for pairwise sequence identities can be shown in the Supplement for the reader’s reference.
  • Included, thank you.
  1. P7, L237: PepMV-Sp13 and PepMV-PS5 isolates, respectively
  • Corrected, thank you.
  1. Fig. 3: The phenotypes of the mutants PepMV-Sp13A113S, and PepMV-Sp13E236Kshould be illustrated in tomato.
  • Unfortunately, we have not tested the PepMV mutants in tomato, and we cannot do it in the short time available to respond to referees. Therefore, we have modified the manuscript to be as specific as possible.
  1. Fig. 5: The locations indicated by boxes should be explained. Organelles such as chloroplasts and nuclei can be indicated.
  • Thank you for your comment. The figure and its legend have been modified accordingly.

Round 2

Reviewer 1 Report

Comments and Suggestions for Authors

The critical points of the manuscript were modified, the paper now deals with the symptom induction on N. benthamiana plants and does not draw conclusions in the case of tomato plants, for which no experiments have been carried out. So I think that the MS can be accepted for publication in it’s the present form .

Author Response

Thank you for your time and interest.

Reviewer 2 Report

Comments and Suggestions for Authors

Although the authors explained that they couldn’t complete the assay of the CP mutants in tomato plants within a short time, I still strongly recommend performing the assay to demonstrate the effects of the CP mutants on the natural host of PepMV, as in N. benthamiana. I would like to ask the editor to give the authors more time to complete the assay. By the way, I don’t see the values for sequence pairwise results in the supplementary file.

Author Response

Dear reviewer,

thank you very much for your observations. We forgot to resubmit the supplementary material for the alignments (Supplementary Table 2), we are very sorry. Now this aspect has been corrected.

Best wishes, Miguel Aranda.